# Encapsidation of Different Plasmonic Gold Nanoparticles by the CCMV CP

**DOI:** 10.3390/molecules25112628

**Published:** 2020-06-05

**Authors:** Ana L. Durán-Meza, Martha I. Escamilla-Ruiz, Xochitl F. Segovia-González, Maria V. Villagrana-Escareño, J. Roger Vega-Acosta, Jaime Ruiz-Garcia

**Affiliations:** Biological Physics Laboratory, Universidad Autónoma de San Luis Potosí, Álvaro Obregón 64, San Luis Potosí 78000, Mexico; analuisa.duranmeza@gmail.com (A.L.D.-M.); iggy_27@hotmail.com (M.I.E.-R.); xochitl.guao@gmail.com (X.F.S.-G.); veronica.villagrana@gmail.com (M.V.V.-E.); rogerveg@mail.ifisica.uaslp.mx (J.R.V.-A.)

**Keywords:** plasmonic nanoparticles, gold nanoparticles, gold nanorods, gold nanoshells, CCMV, virus-like particles

## Abstract

Different types of gold nanoparticles have been synthesized that show great potential in medical applications such as medical imaging, bio-analytical sensing and photothermal cancer therapy. However, their stability, polydispersity and biocompatibility are major issues of concern. For example, the synthesis of gold nanorods, obtained through the elongated micelle process, produce them with a high positive surface charge that is cytotoxic, while gold nanoshells are unstable and break down in a few weeks due to the Ostwald ripening process. In this work, we report the self-assembly of the capsid protein (CP) of cowpea chlorotic mottle virus (CCMV) around spherical gold nanoparticles, gold nanorods and gold nanoshells to form virus-like particles (VLPs). All gold nanoparticles were synthesized or treated to give them a negative surface charge, so they can interact with the positive N-terminus of the CP leading to the formation of the VLPs. To induce the protein self-assembly around the negative gold nanoparticles, we use different pH and ionic strength conditions determined from a CP phase diagram. The encapsidation with the viral CP will provide the nanoparticles better biocompatibility, stability, monodispersity and a new biological substrate on which can be introduced ligands toward specific cells, broadening the possibilities for medical applications.

## 1. Introduction

Most viruses protect their genomic material through a protein shell called a capsid. The virus capsid has various interesting properties such as monodispersity, stability and, in most cases, the capsid self-assembles around its genomic material spontaneously to form a virion. Furthermore, it has been shown that the capsids of some viruses can self-assemble around negative cores that mimic its genomic material [1]. Moreover, it has also been shown that under the appropriate conditions, capsids can self-assemble without a genome to form empty capsids [2]. Therefore, viral capsids could be used as nanocontainers for various synthetic or biological materials. However, for the exogenous material to be encapsidated, must have similar characteristics to the of native viral genetic material, such as charge and size [3].

Cowpea chlorotic mottle virus (CCMV) is an icosahedral plant virus that infects *Vigna Unguiculata* [4]. The capsid is made of 180 identical proteins and has a so-called triangulation number, T = 3. At pH < 6 it has an external diameter of ~28 nm and an internal diameter of ~21 nm [5]. At higher pH values, the capsid increases its size as it swells [6]. CCMV was the first icosahedral virus to be disassembled and reassembled in vitro, when their subunits were put together in an appropriate solution [6], and it has been proven that other similar viruses can also self-assemble in a similar way [7]. In most cases, the self-assembly process happens spontaneously, just by mixing the purified capsid protein (CP) with its genetic material. The interactions that lead to the virion self-assembly formation are electrostatic in nature, between the interior part of the CP and its genome; since the genomes in solution are negatively charged and the CP have a positive N-terminus that interacts with the genome [8]. CCMV is easy to amplify [4] and there are well known methods for its disassembly, protein purification and reassembly [6] with and without RNA [7]. Furthermore, the CP has been shown to be biocompatible [9], which makes the CCMV capsid an excellent candidate for its use as a nanocontainer for biological and medical applications.

Remarkably, the purified CP of CCMV, under certain conditions, can reassemble not only into empty capsids, but also form other structures such as tubes, disks and multi-wall capsids [10,11]. These types of assemblies have been studied under a variety of different conditions of ionic strength (I) and pH [11], temperature, protein N-terminus deletion effect [12], and protein concentration [8]. It turns out that the isoelectric point (pI) of both the capsid and the CP are of capital importance for the understanding self-assembly; a ionic strength (I) vs. pH phase diagram of the CCMV CP has been constructed [11], which has helped us to understand the self-assembly behavior of the CCMV CP in the absence of its genetic material. Various capsid polymorphs have been identified by electron microscopy, as it is shown schematically in Figure 1 [8,11], where four main regions, corresponding to the formation of icosahedral T = 3 capsids (single capsids), multiwall capsids (one capsid on top of another), tubes and disassembled proteins can be observed. This diagram help us to choose the appropriate pH and I to generate virus-like particles (VLPs) for the different gold nanoparticle shapes; the conditions for each assembly are schematically shown as yellow-colored structures in Figure 1, corresponding to the shapes of the gold nanoparticles used.

Several groups have used the CCMV CP to encapsidate different nuclei such as negatively charged nanoparticles of TiO_2_ [13], negatively charged polymers such as polystyrene sulfonate (PSS) [14], Prussian blue nanoparticles [15], nucleic acid different from the native CCMV RNA [16] and other negatively charged particles [17]. In all cases, these VLPs formation are driven by electrostatic interactions mediated by pH and I.

On the other hand, gold nanoparticles have been studied due to their unique optical and electronic properties, and have been the subject of substantial research, with applications in a wide variety of areas, including electronics, nanotechnology, and biomedicine and the so-called area of nanomedicine [18]. The properties of these nanoparticles, and therefore their applications, depend strongly on their size and shape [19]. Different shapes of gold nanoparticles have been synthetized such as nanospheres [20], nanorods [21], nanostars [22], nanotubes [23], nanocubes [24], nanodisks, nanowires [25], and nanoshells [26]. One of the main features of these nanoparticles is that their absorption peak maximum depends on their morphology, which is not only of scientific but also of technological interest. For example, gold nanorods and nanoshells have their absorption peak located in the transparent window of biological tissues—that is, in the near infrared region (NIR), from about 750 to 1200 nm [27]. For such particles, their strong surface plasmon absorption can convert NIR radiation into heat [28], which is favorable for their use in photothermal therapies for cancer treatment [29], photothermally triggered drug release [30], and gene therapy [31]. In addition, their strong scattering also makes them good contrast agents for medical imaging [32].

Various groups have encapsidated solid spherical gold nanoparticles with viral capsids, for example, solid AuNPs with sizes ranging from 5 to 15 nm have been encapsidated with the red clover necrotic mosaic CP [33]. Dragnea’s group has also conducted extensive research on packaging solid spherical AuNPs of different sizes. They reported the self-assembly of viral protein from the brome mosaic virus (BMV) around citrate-coated gold nanoparticles of about 2.5–4.5 nm in diameter in the absence of its nucleic acid, which formed a capsid around the particles, where the interaction forces that create these VLPs are purely electrostatic [1]. In a different study they use 12 nm AuNPs, functionalized with a layer of the hydrophilic molecule triethylene glycol (TEG). After functionalization of the particles, the diameter of the construct increased to about 16 nm, which approximates the inner diameter of the capsid of the BMV virus. The size of the VLPs obtained was 26 ± 2 nm in diameter, which is identical in size to the native virus [34]. They conducted a study on the self-assembly of BMV CP around spherical gold nanoparticles of 6, 9, 12, 15 and 18 nm after being functionalized with polyethylene glycol (PEG), which is also a hydrophilic polymer, and successfully formed VLPs, they used a ratio of 1:270 AuNP:proteins and reported that the self-assembly efficiency is a function of the gold core diameter and reaches a maximum with the functionalized cores of 16 nm, which is about the size of the interior diameter of the native virus [35]. They studied the encapsidation of AuNPs of 6 and 12 nm in diameter coated with thiol-alkylated tetra (ethylene glycol) (TTEG), with CCMV; however, in this work they used a mutant of CCMV CP lacking most of the N-terminal domain. They concluded that is possible to encapsidate AuNP with a cleaved protein lacking its amino terminal, which results in an increased flexibility in the dihedral angle of the dimer, resulting in capsids that can vary in size and structure capable of encapsidating different core sizes [36].

In order to increase gold nanoparticles’ biocompatibility, monodispersity and enhance their surface characteristics, on which a ligand can be added to provide specificity to the nanoparticles, we propose the encapsidation of gold nanoparticles with the purified native CP of the CCMV virus. Therefore, we present a procedure for the encapsidation of different types of gold nanoparticles: solid spherical gold nanoparticles (AuNPs), ultrasmall gold nanoshells (AuNSs) and gold nanorods (AuNRs). Even though functionalized gold nanoparticles with Bis-p-(sufonatophenyl)phenyl Phosphine (BSPP) ligands have been encapsidated with CCMV [37], we show for the first time the direct use of wild-type CCMV CPs, not only for the encapsidation of spherically shaped gold nanoparticles but also rod-shaped nanoparticles without ligands. Gold nanoshells and gold nanorods have advantages and disadvantages for medical applications—both can be used for photothermal therapy and imaging contrast. However, gold nanoshells are highly unstable due to the Ostwald ripening process [38]. While the synthesis of gold nanorods produce particles with a highly positive surface charge, which is highly cytotoxic [39]. These gold nanoparticles have shown great potential in medical applications and their encapsidation can not only improve their biocompatibility, but also increase their lifetime—for example, by enhancing the biocompatibility in the case of AuNRs or improving the lifetime of AuNSs, as well as producing monodisperse VLPs, and creating a new substrate around the nanoparticle on which a ligand can be attached to produce VLPs with desirable qualities [40].

## 2. Results

### 2.1. Synthesis of Gold Nanoparticles

Typical images of the different synthetized gold nanoparticles were observed with transmission electron microscopy (TEM). The morphology of both the 5 and 20 nm-diameter nanospheres, with nanoparticles having a quasi-spherical shape, are shown in Figure 2a,b. Gold nanorods are shown in Figure 2c, where we can observe that they also have a regular morphology. Figure 2d shows an image of the gold nanoshells.

### 2.2. Absorption Spectra of Gold Nanoparticles

The AuNP, AuNR and AuNS absorption spectra are shown in Figure 3. The AuNPs exhibit a characteristic absorption peak for their size (15.6 ± 2.9 nm by TEM or 18 ± 0.3 nm of hydrodynamic diameter by DLS, see Appendix A) in the visible region, at about 525 nm. Both the AuNRs and the AuNSs show an absorption peak in the NIR region: The AuNRs show a broad peak with a maximum at about 780 nm, while the AuNSs show a more defined peak with a maximum at about 815 nm.

### 2.3. Encapsidation of AuNP

AuNPs of 18 ± 0.3 nm of hydrodynamic diameter (see Appendix A) with a surface charge of −49 mV (obtained by dynamic light scattering, not shown) were successfully encapsidated by the CCMV CP, as can be seen in Figure 4 (negative stained TEM). All the AuNPs have a capsid, at pH = 4 and I = 0.1 M, and with the correct molarity of protein; this was expected since the particles have a highly negatively surface charge, which interact with the positively charged N-terminus of the protein, leading to a 100% encapsidation. Under these conditions, empty capsids also form, because of the excess of CCMV CP we can observe several empty capsids. At pH = 7 and I = 0.1 M the encapsidation is also highly efficient due to the fact that the negative charge of the AuNPs is also quite high, but no empty capsid can be seen, since at these conditions, tubes are the equilibrium structures; however, a very high concentration of CCMV CP is required to form empty tubes [8]. In this case, we conclude that the spherical shape of the nanoparticles forces the CP to self-assemble into spherical capsids around the nanoparticles (Figure 4b).

On the other hand, several 5 nm AuNPs can be encapsidated in a single capsid, as shown in Figure 5, at pH = 4 and I = 0.1 M. It can be observed in Figure 5a that the capsids can contain from one up to about 10 nanoparticles. Moreover, a high number of nanoparticles can be trapped in a jointed capsid structure, as shown in Figure 5d, where it can be seen that three capsids surround the AuNP, similar to those found in the formation of VLPs, with very long RNA cores [16].

### 2.4. Encapsidartion of AuNSs

The synthetized AuNS have an average diameter of 25 ± 5 nm and an average surface charge average of −22 mV at pH 4. At this pH, the CCMV CP total charge is slightly negative [8], but still the arginine tails are expected to interact with the AuNSs; therefore, at these conditions, it was expected that the CCMV CP and the AuNSs would interact electrostatically. Note that the selected conditions for the encapsidation are in the limit of single capsid and multiwall formation. We chose this condition due to many AuNSs having a diameter greater than the inner diameter of a single capsid. In this way, we wanted to be sure that the AuNS would be completely covered by the CCMV CP. As we can see in Figure 6 (and S4), all the AuNS were encapsidated, although due to the experimental conditions of the encapsidation, and due to the excess of CP, we can also observe the formation of empty capsids of different sizes (see Figure 6c).

As mentioned above, the diameter of the AuNSs exceed the 21 nm inner diameter of the native virus; because of this, the capsids formed around most of the AuNS are multiwall capsids, creating VLPs with average diameter of 43.9 ± 6.4 nm (see Appendix A). The CCMV CP is forced to self-assemble into spherical shells that are not perfect icosahedra [8,11]. Consequently, the distribution of protein conformations on larger particles is significantly different than in the wild-type capsid. This has also been observed in the encapsidation of large nanoemulsion droplets with the CCMV CP [41]. 

### 2.5. Encapsidation of AuNRs

The AuNRs synthetized for this work are 48.75 ± 8.04 nm long and 12.73 ± 2.36 nm wide, with an aspect ratio of 3.84 and a maximum absorbance peak at ~780 nm (see Figure 3). After the synthesis process, the AuNRs have an average zeta potential of +20 mV, determined by dynamic light scattering. Figure 7a The treatment with l-α-phosphatidic acid (PA) changed the surface charge of the AuNRs from positive to negative, to an average zeta potential value of −63.8 mV Figure 7a. Figure 7b,c show TEM images of the AuNRs before and after treatment with PA. 

After mixing the AuNRs with the CCMV CP, we found that the AuNRs are encapsidated in several ways, as shown in Figure 8 (see also Appendix A); one way in which nanorods are contained in a single nonsymmetrical capsid as can be seen in Figure 8a, while in most nanorods, the proteins form a nanotube capsid along the longer axis, wrapping them up, as can be seen in Figure 8b. However, in some nanorods, the CP form icosahedral structures on the tips of the nanorods, as is shown in Figure 8c.

## 3. Discussion

Here, we show the formation of VLPs that contain plasmonic gold nanoparticles of different shapes as their core, such as gold nanospheres, nanorods and nanoshells. These nanoparticles have physical properties that are of interest in nanomedicine, e.g., their capacity for use as imaging contrast agents and in photothermal therapy.

Since the quasi-spherical AuNPs have a highly negatively surface charge, they interact with the positively charged *N*-terminus of the protein leading to a 100% encapsidation. At low pH, empty capsids are also formed, therefore in this condition due to the excess of CCMV CP, we can observe several empty capsids. However, at pH = 7 and I = 0.1 M the encapsidation is also highly efficient. This was surprising because at this condition the formation of protein nanotubes is the preferred self-assembly structure. In addition, at these conditions, a high concentration of protein is needed to form any self-assembly structure [8,11]. In this case, we conclude that the spherical shape and the high negative charge of the nanoparticles forces the CP to self-assemble into spherical capsids around them. In addition, at these conditions, no empty capsid can be observed, since at these conditions protein nanotubes are the equilibrium structures, but a very high concentration of CCMV CP is required to form empty tubes [8]. In the case of the small quasi-spherical AuNPs, we found that a different number of particles can be encapsidated. Moreover, sometimes we observed the formation of jointed capsids (several capsids sharing the same nucleus) that contained many these small nanoparticles.

The AuNSs are very interesting nanoparticles, since they absorb in the NIR region in a similar manner to the AuNRs, and they have a great potential to be used in photothermal therapy. The experimental conditions in which the AuNSs were encapsidated were selected to try to avoid partial encapsidation, especially for the larger nanoparticles. However, we found that all AuNSs were encapsidated by two to three capsids; this conclusion was determined from the thickness of the protein layer around the AuNSs, which is ~10 nm, and in the wild-type virus is about 3.5 nm [42]. In the case of the AuNSs, even though their average diameter might be greater that the inner diameter of the native virus, the electrostatic interaction between the CCMV CP with the gold nanoparticles is strong enough to follow the shape and size of the nanoparticle. However, it is known that the CCMV capsid is quite flexible and it is capable of encapsidating larger genomes than its natural genome [16], and the larger AuNSs might be a similar case. A drawback for the use of AuNSs is their well-known instability, because in short periods of time they become fragmented due to the Ostwald ripening process [38]; Ostwald ripening leads to the dissolution of smaller nanoparticles into larger ones through coalescence in order to decrease the interfacial energy between the nanoparticles and the solvent. Colloidal nanoparticles become increasingly unstable with decreasing particle size, due to the relative increase in surface energy. The solubility concentration of the particle is directly proportional to the surface tension and inversely proportional to the radius of the nanoparticle. Therefore, the AuNSs are destroyed via the transfer of gold atoms from smaller shells to larger ones. Furthermore, gold nanoshells are susceptible to photothermal fragmentation when exposed to high IR incident intensities [43]. It has been shown that encapsidation by the CCMV capsid can provide stability against Ostwald ripening in nanoliposphere systems [44]. Therefore, the encapsidation could provide not only a biocompatible surface, but also an envelope that can give them a better stability against fragmentation either due to Ostwald ripening or due to photofragmentation.

On the other hand, AuNRs are nanoparticles best known for their use in photothermal therapy applications [45]. As the other gold nanoparticles in thesolution, these nanorods need to be highly charged to avoid aggregation. One of the most known synthesis methods produces them with a very high positive surface charge. It has been shown that particles with a high positive charge are undesirable because they are cytotoxic. Therefore, we developed a method to change the surface charge of the AuNRs from positive to negative charge, to make them more biocompatible. Even though the encapsidation of the AuNRs was also successful, the shapes of the capsids around them were not uniform, as in the case of the AuNPs and AuNSs. 

In general, we demonstrate that it is possible to encapsidate different types of gold nanoparticles with the CP of the plant virus CCMV, without the surface modification of the nanoparticles, as has been reported [35,36], with the only surface charge change taking place in the case of the AuNRs. We varied the experimental conditions—that is, we varied the pH, ionic strength and concentration of the CCMV CP conditions; the concentration of the CCMV CP was a very important parameter to obtain an encapsidation efficiency of 100% in all cases.

## 4. Materials and Methods 

### 4.1. Materials

Silver nitrate (AgNO_3_; 99.99%), sodium borohydride (NaBH_4_; 99%), trisodium citrate di-hydrated (NaCit; 99%), potassium carbonate (K_2_CO_3_; 99%), cloroauric acid (HAuCl_4_; 99.9%), hexadecyltrimethylammonium bromide (C_19_H_42_BrN; 99%), hydrogen peroxide (H_2_O_2_; 30%) Phenylmethylsulfonylfluoride (PMSF; 99%) calcium chloride (CaCl_2_; 99%), citric acid (C_6_H_8_O_7_; 99.5%), dithiothreitol (DTT; 99%), Tris(hydroxymethyl)aminomethane (Tris; 99.8%) (Sigma Aldrich, Saint Louis, MO, USA), sodium hydroxide (NaOH; 97%) (Fisher BioReagents, Waltham, MA, USA), sodium chloride (NaCl; 100%) (JT Baker, Phillipsburg, NJ, USA), ethylenediaminetetraacetic acid (EDTA; 99%, Bio Rad, Hercules, CA, USA), l-α-phosphatidic acid (PA; 99%) (Avanti Polar Lipids, Alabaster, AL, USA). All materials were used without further purification. Stock solutions of sodium borohydride, sodium citrate, and silver nitrate were freshly prepared for each reaction. Deionized bioresearch grade and sterilized milliQ-type water was used for all reactions.

### 4.2. Synthesis of AuNP

The spherical gold nanoparticles were synthetized by the inverted Turkevich method [46], which consists of a reduction of gold salt HAuCl_4_, with sodium citrate as a reduction agent at its boiling temperature. In order to obtain the desired size of 18 nm, we used a molar ratio of 3.4 M HAuCl_4_:sodium citrate and, for the 5 nm particles, we used a molar ratio of 20 M HAuCl_4_:sodium citrate; this procedure produced highly monodispersed particles.

### 4.3. Synthesis of AuNSs

For the synthesis of AuNSs, we used a combination of the seed-mediated growth using the Lee–Meisel method, which consists of a thermal reduction with citrate [47]. Seed solution: 2 mL of 1% NaCit was mixed with 7.6 mL of water and vigorously stirred while heating at 70 °C for 15 min. Then, 170 µL of AgNO_3_ 1% and 200 µL at 4 °C of NaBH_4_ 0.1% solution were added at the same time, while keeping the temperature constant and stirring for 1 h, obtaining 4 nm nanoparticles in diameter. Then, we grew the AgNPs. In a round-bottom flask, 400 µL of NaCit 1% solution and 16 mL of water were mixed with stirring and 100 °C for 15 min. Then, 2 mL of the seed solution and 340 µL of AgNO_3_ 1% solution were added, while stirring and heating continued for 60 more min; the resulting solution was kept at room temperature in darkness. After that, the gold layer was formed onto the AgNP by deposition in a procedure adapted from Vongsavat et al. [48], and improved by Duran-Meza et al. [47] that involves the preparation of a potassium–gold (K–gold) solution and then the mixing of the AgNPs with this solution. Briefly, in a round bottom flask protected from light, 10 mL of deionized water and 2.5 mg of K_2_CO_3_ were added. Then, 200 µL of 1% solution of HAuCl_4_ was incorporated to the mixture under vigorous stirring at room temperature for 30 min. The resulting solution was kept for 12 h at 4 °C before use. However, we discovered that the thickness of the gold shell can be modified by varying the reaction time and/or the volume ratio of the K–gold to the AgNP solutions [39]: The K–gold solution was added into the AgNPs solution at a volume ratio of 1:1 to obtain a 25 ± 5 nm AuNSs with a 4 nm thickness that have a maximum absorption peak in the near infrared range at 815 nm. To clean the AuNSs, the resulting solution was centrifuged at 6000 g (Biofuge Stratus, Heraus, Germany) for 10 min, and the sediment was resuspended in water. The centrifugation and resuspension process was repeated three times and the solution was kept at room temperature.

### 4.4. Synthesis of AuNRs

We prepare AuNRs by the method developed by Xu et al. [49] that produces a well-controlled morphology, high purity, and good quality monodisperse rods. This method involves the preparation of a gold seed solution and the growing of the AuNRs at alkaline conditions. Briefly, a seed solution was prepared by adding HAuCl_4_ to a Cetyltrimethylammonium Bromide (CTAB), solution, and then NaBH_4_ while stirring for 2 min, to a final concentration of 5 × 10^−3^ M HAuCl_4_, 0.2 M CTAB and 0.01 M NaBH_4_, then the solution was aged for at least 2 h before use at room temperature. Then AuNRs were grown and functionalized by using 10 mL of a solution composed of 5 × 10^−3^ M HAuCl_4_, 0.1 M CTAB, 5 × 10^−3^ M NaOH, 0.02 M H_2_O_2_, 0.06 nM AgNO_3_ and 15 µL of the seed solution. The resulting solution was centrifuged at 6000 g for 10 min and resuspended in water. The process was repeated and the solution was kept at room temperature. AuNRs prepared by this procedure usually present an absorbance peak in the NIR region, which can be compared in their thermal absorbance with the AuNSs [47]. AuNRs synthetized by this method produce highly positively charged AuNRs due to the stabilizing surfactant bilayer made out of CTAB. However, a highly positively charged nanorod surface is undesirable because it is cytotoxic [39]. In addition, it will not interact appropriately with the CCMV CP, because the core surface must be negatively charged; therefore, the AuNRs require a change in the superficial charge from positive to negative, so the positively charged *N*-terminus of the CCMV CP can interact with the negatively charged surface of the AuNRs and self-assemble around them. In order to make this surface charge change, we used a volume ratio 1:1 of 0.02 M AuNRs solution and 1 M PA and sonicated this solution for 10 min. The solution was stored at 4 °C for 24 h. The PA is a biocompatible anionic lipid, which forms a lipid bilayer on the AuNPs, replacing the CTAB on the nanorod surface, since CTAB is not covalently attached to the nanorod surface, thus reversing the nanorods’ surface charge. The final superficial change, from positive to negative, was monitored by micro electrophoresis with Dynamic Light Scattering (DLS) in a Zetasizer Nano ZS (Malvern Instruments, Worcestershire, England).

### 4.5. CCMV CP Purification

CCMV was harvested and purified from infected Cowpea plants (*Vigna ungiculata*) following established protocols [4,50]. Briefly, it was dialyzed against disassembly buffer (500 mM CaCl_2_, 50 mM Tris-HCl pH = 7.5, 1 mM EDTA, 1 mM DTT, and 0.5 mM PMSF) for 24 h in order to disrupt the viral capsids and separate the CP and the RNA. Then the disassembled virus was ultracentrifuged, at 536,600 g using a TLA110 rotor for 106 min at 4 °C, in order to precipitate the RNA and recover the purified protein still in suspension. The recovered protein was then dialyzed in protein buffer (1 M NaCl, 50 mM Tris-HCl pH = 7.2, 1 mM EDTA, 1 mM DTT, 1 Mm PMSF) that stabilized the protein into dimers [51]. Protein concentration and purity was measured in a UV-Vis spectrophotometer (NanoDrop 2000c UV-Vis Spectrophotometer, Thermo Fisher Scientific Inc., Waltham, MA, USA), and only the fractions with 280/260 ratios above 1.5 were chosen and used for the encapsidation experiments. 

### 4.6. AuNPs Encapsidation

Capsid formation around the nanoparticles was performed during 24 hr microdialysis for each of the gold nanoparticles used at each particular condition shown in Figure 1. AuNP of 18 nm were encapsidated in three different conditions, citrate buffer (citric acid/sodium citrate) pH = 4, I = 0.02 M, citrate buffer pH = 3.6, I = 0.1 M and phosphate buffer (sodium phosphate monobasic/sodium phosphate dibasic) pH = 7, I = 0.1 M. In all cases, the ratio of 540 proteins per particle was used, favoring the formation of icosahedral capsids. The AuNPs of 5 nm were encapsidated in citrate buffer pH = 4 and I = 0.1 M, and at a ratio of 540 protein per five particles [11]. AuNSs were encapsidated in citrate buffer at pH = 4 and I = 0.01 M, and a ratio of 540 proteins per particle was used again. In order to promote the formation of protein tubes around the AuNRs, a phosphate buffer at pH = 6.5, I = 0.01 M and a ratio of 960 proteins per particle was used.

### 4.7. Characterization Methods

The size and morphology of the AuNP, AuNSs, and AuNRs were characterized by transmission electron microscopy (TEM), using a JEOL 1230 instrument, at 100 kV. The samples were prepared by depositing an 8 µL drop onto a copper grid (400 mesh) that was previously coated with parlodium and carbon. After 1 min, the excess sample was removed with a filter paper and then it was negatively stained with 6 µL of a 2% uranyl acetate solution for 60 s and dried again; the grid was stored in a desiccator overnight. The images were processed using the ImageJ program [52], the size distribution histograms were constructed from the analysis of at least 200 particles, and each nanoparticle was measured in two directions. The absorption extinction spectra curves were measured by UV-Vis and IR Spectroscopy at room temperature on a Spectrophotometer UV-Vis (Nanodrop 2000, Thermo Scientific (Waltham, MA, USA). Zeta potential was measured by electrophoresis in a Zetasizer Nano ZS. 

## 5. Conclusions

Recent advances in nanotechnology have resulted in the development of different types of metal nanoparticles for biomedical applications; among them, gold nanoparticles stand out for the ease with which they are synthesized, their size and tunable optical properties [53]. However, there is great interest in giving these nanoparticles more specificity toward the cells they bind to [54,55]. We report the formation of VLPs that contain plasmonic gold nanoparticles of different shapes and physical properties that are of interest in nanomedicine. Among them are gold nanospheres, nanorods and nanoshells. It is important to mention that it was possible to encapsidate these types of gold nanoparticles with the complete CP of CCMV and without the need of capping the particles with TEG or PEG [35,36], under different pH, ionic strength and concentrations of the CCMV CP conditions. The presence of a negatively charged core, such as the AuNPs, the CP interaction with the core is strong enough to drive the self-assembly of the CP around it to form a spherical VLP at this condition. This result is surprising for two reasons; first, at this high pH and I, the concentration of protein required for self-assembly into empty structures is much higher and, second, at these conditions, the preferred self-assembled structure of the CP is tubular. Therefore, our results show that the interaction of the CCMV CP with a negative core is strong enough to drive its self-assembly into the shape of the core.

In the case of the AuNSs, even though their average diameter might be greater that the inner diameter of the native virus, the electrostatic interaction between the CCMV CP with the gold nanoparticles is strong enough to follow the shape and size of the nanoparticle. However, it is known that the CCMV capsid is quite flexible and it is capable of encapsidating larger genomes than its natural genome [16]. In addition, encapsidating them in the conditions under which multiwall capsids are formed ensures that the AuNSs are completely surrounded by CP, which lead to an encapsidation efficiency of 100% of the AuNSs. The VLPs formed with this procedure have a mean diameter of 43.9 nm, indicating the formation of two to three capsids. Furthermore, the encapsidation might help to stabilize the AuNSs against fragmentation due to either Ostwald ripening or photofragmentation [38,43]. Because of the excess CCMV CP used, empty capsids are also formed The VLPs formed with the AuNRs are encapsidated in two different manners; in some cases, the capsid form an oval shape, but most of the nanorods are encapsidated in protein tubular structures and, in some cases, the tubular capsids around the nanorods show empty spherical capsids at the tips of the tubes. It is important to remark that the AuNRs require a change in their charge due to the synthesis method produce them with a high positive surface charge, which is cytotoxic. Here, we show a procedure to reverse the surface charge of the nanorods from a positive to a negative charge, which makes them more biocompatible.

In summary, we show that it is possible to encapsidate these types of gold nanoparticles with the CP of the plant virus CCMV, without surface modification of the nanoparticles, with the exception of the gold nanorods, where the surface charge was changed for the first time in these types of systems, from positive to negative lipid layers. In addition, we varied the experimental conditions—that is, we varied the pH, ionic strength and concentration of the CCMV CP conditions, and found a 100% encapsidation efficiency in all cases. Moreover, it is known that the CCMV CP can be easily functionalized to give to the VLPs a more specific cell targeting ability [56]. It is also important to emphasize that these nanoparticles are nontoxic for use in in vivo therapies [57], and that it has been proved that VLPs based on the CCMV CP can directly transfect cancer mammalian cells [58], which makes them very attractive systems. 

## Figures and Tables

**Figure 1 molecules-25-02628-f001:**
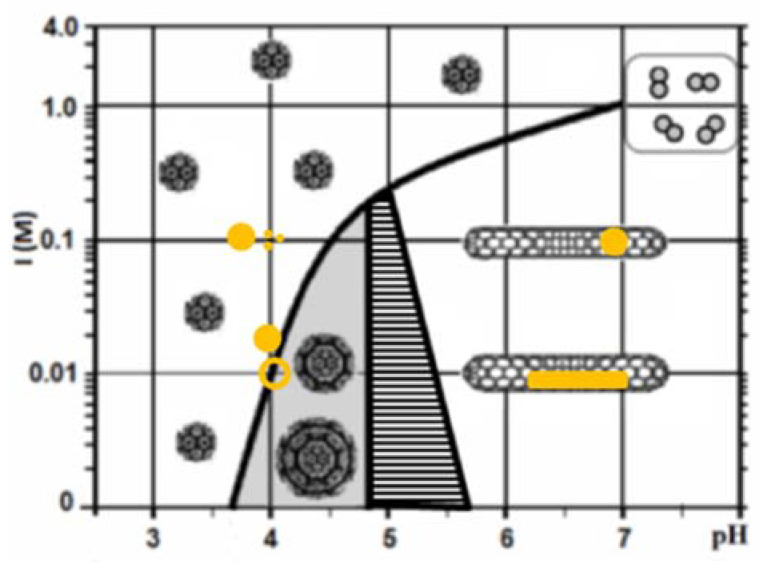
Phase diagram of the cowpea chlorotic mottle virus (CCMV) capsid protein (CP) self-assembly as a function of pH and ionic strength. The golden color structures represent each of the different gold nanoparticles; gold nanospheres (Ø ~ 18 and ~5 nm in diameter), gold nanoshells (Ø ~ 25 nm) and gold nanorods (aspect ratio ~4.1 nm). Each structure is placed under the conditions in which the virus-like particle (VLP) assemblies were formed. The figure was adapted from [8].

**Figure 2 molecules-25-02628-f002:**
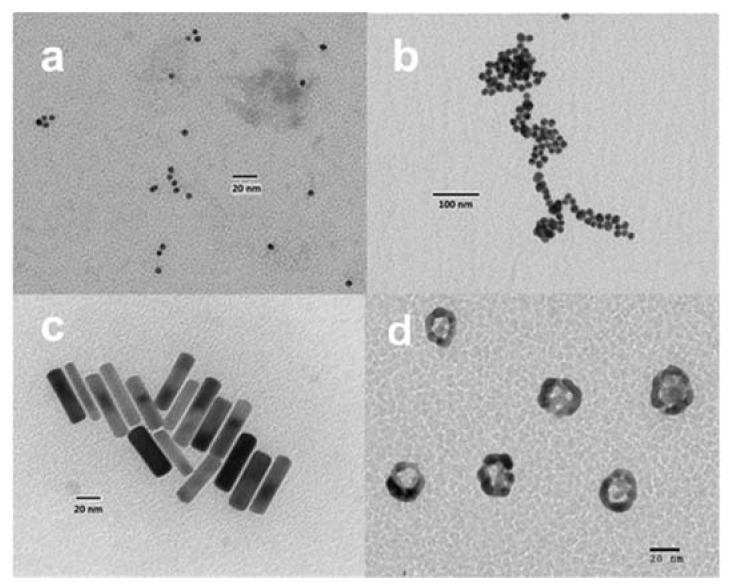
Negative stained transmission electron microscopy (TEM) micrographs of typical images of the diferent gold synthesized nanoparticles, (**a**) and (**b**) 5 nm and 18 nm of solid spherical gold nanoparticles (AuNPs), (**c**) gold nanorods (AuNRs), and (**d**) ultrasmall gold nanoshells (AuNSs).

**Figure 3 molecules-25-02628-f003:**
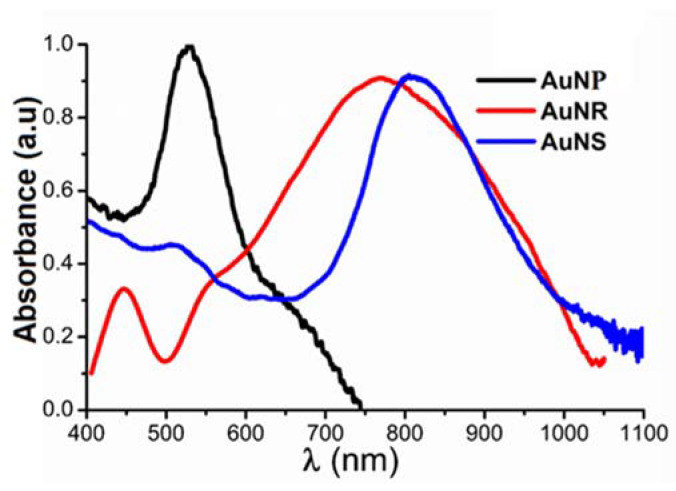
Typical UV-Vis-near infrared region (NIR) absorption spectra of AuNPs (18 nm), AuNR and AuNS. Note that the spectra of the AuNR and AuNS peak at the near infrared region, around 800 nm.

**Figure 4 molecules-25-02628-f004:**
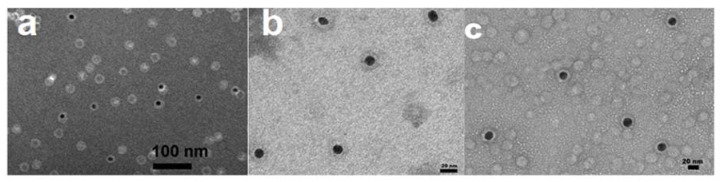
Negative stained TEM micrographs of 18 nm AuNP encapsidated by the CCMV CP at different conditions (**a**) pH = 3.6 and I = 0.1 M. (**b**) pH = 7 and I = 0.1 M and (**c**) pH = 4 and I = 0.02 M.

**Figure 5 molecules-25-02628-f005:**
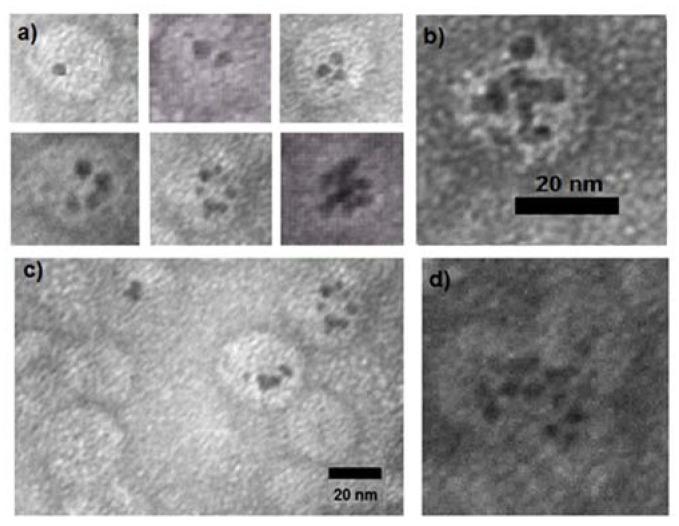
Negative stained TEM micrographs of 5 nm AuNPs encapsidated by the CCMV CP at pH = 4 and I = 0.1 M. The black dots are the AuNPs. (**a**) and (**b**) Single capsids containing from 1 to 10 AuNPs. (**c**) General image showing empty capsids and single capsids containing several AuNps. (**d**) A high number of nanoparticles trapped in a jointed capsid structure.

**Figure 6 molecules-25-02628-f006:**
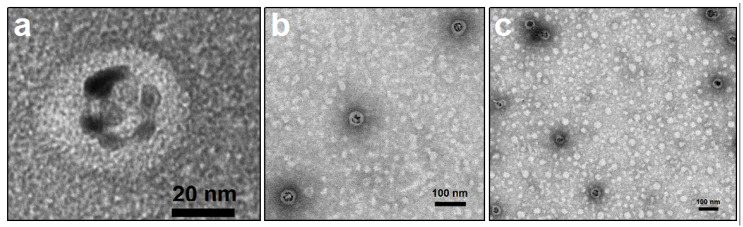
Negative staineded TEM micrographs of AuNSs encapsidated by CCMV CP at pH 4 and I = 0.01 M (**a**) and (**b**) closer look of AuNSs encapsidated with CCMV CP. (**c**) Wider image AuNSs encapsidated with CCMV CP were it can also be seen empty capsids (clearer spheres with no core).

**Figure 7 molecules-25-02628-f007:**
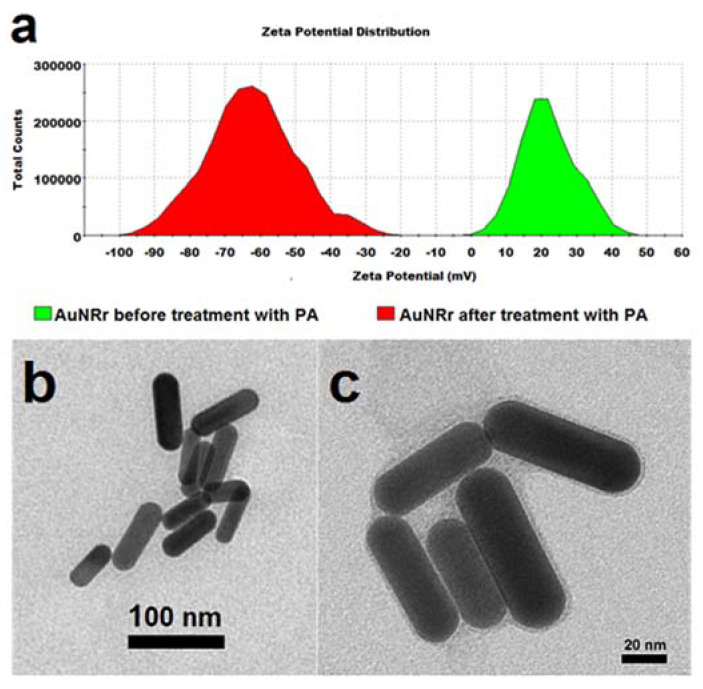
(**a**). Zeta potential measurements of AuNRs before (green) and after (red) treatment with l-α-phosphatidic acid (PA). (**b**) Negative stained TEM of AuNRs after synthesis. (**c**) AuNRs after treatment with PA.

**Figure 8 molecules-25-02628-f008:**
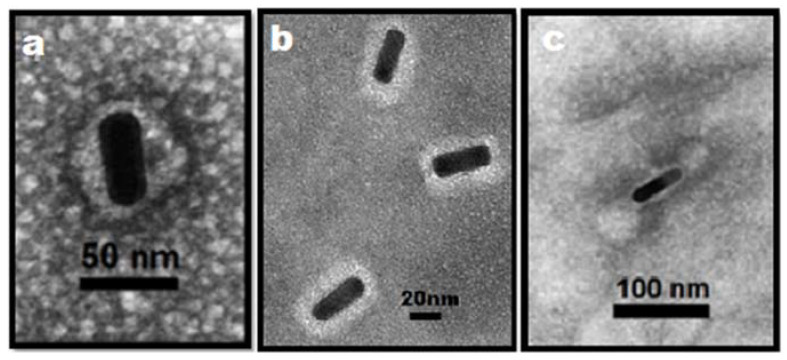
Negative stained TEM micrograph of gold nanorods encapsidated by the CP of CCMV, forming different enveloped structures. (**a**) Nanorods are contained in a single nonsymmetrical capsid. (**b**) The proteins form a nanotube capsid along the longer axis. (**c**) In addition to encapsidating the nanorod, the CP forms icosahedral structures on the tips of the nanorods.

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
