# Peer review of "Encapsidation of Different Plasmonic Gold Nanoparticles by the CCMV CP"

_molecules, 2020, doi:10.3390/molecules25112628_

Round 1

Reviewer 1 Report

This manuscript shows the encapsidation of various gold nanoparticles using cowpea chlorotic mottle virus (CCMV) capsid protein. This study is interesting to researchers on the fields of nano-science and nano-technology and applicable to medical researches such as cell imaging, bio-sensing and photothermal cancer therapy. However, this manuscript has some doubts to be solved and is not acceptable in this journal. Some comments are described below.

  1. p2, line 50, a space should be added between “genome” and “[8]”.
  2. In the “Introduction”, the authors use “capsid” and “capsid protein” What is its difference? The authors should describe it in the text clearly.
  3. The “Introduction” and “Conclusion” are too long.
  4. p4, line 161, “nm” in front of “20 nm” should be deleted.
  5. In Figure 4b, the picture is not clear compared to Figure 4a and 4c. The picture should be replaced with new one.
  6. p5, line 180-181, the authors mention 100% of the encapsidation. The number of counted VLPs in each experiment should be shown as a data.
  7. p6, line 202, what is the difference between “single capsid” and “multicapsid”.
  8. p6, line 212, the authors mention most of the AuNS are multishell capsid creating VLPs. How did the authors reveal the multishell capsid structure.
  9. The authors should show the data proving the encapsidation of the capsid protein by immunoTEM or ELISA etc.
  10. It is doubtful whether the capsid protein formed VLPs or aggregated on the surface of gold nanoparticles.
  11. p8, line 251, what is “jointed multicapsids”?
  12. p8, line 256-257, how the authors reveal the encapsidation of AuNSs by two to three protein shells?
  13. p8, line 292, “K2CO3” is not correct.
  14. p10, line 362, “rpm” should not be used.

Author Response

Response to Reviewer 1 Comments

  1. p2, line 50, a space should be added between “genome” and “[8]”.

ANSWER: We appreciate the correction, the space was added

  1. In the “Introduction”, the authors use “capsid” and “capsid protein” What is its difference? The authors should describe it in the text clearly.

ANSWER: We appreciate the suggestion in fact it was confusing to the reader so we explained in line 31 in the new version what we refer as a capsid, and in line 47 what a capsid protein is, and we use CP for capsid protein in the rest of the text.

  1. The “Introduction” and “Conclusion” are too long.

ANSWER: We appreciate the comment, we indeed reduce the introduction by deleting the paragraph from lines 72 to 91 where we explain Figure 1 since it was too repetitive, also, we resume the background from lines 113 to 141 in the old version to line 93 to 113 in the new version and we also did some minor changes complete introduction to make it shorter.

In the other hand, we only made minor changes to the conclusion because we think the information is important to understand the relevance of the paper.

  1. p4, line 161, “nm” in front of “20 nm” should be deleted.

ANSWER: Thank you for noticing it, it was deleted after 5 nm  “…both the 5 and 20 nm diameter nanospheres”

  1. In Figure 4b, the picture is not clear compared to Figure 4a and 4c. The picture should be replaced with new one.

ANSWER: We appreciate the comment, Figure 4b do not have the same contrast as figures 4a and 4c because at that pH, as it can be seen in the phase diagram, it is not possible to have empty capsids, so the excess capsid protein CP, that do not interact with the particle is in the background reducing the contrast, but we think that it is still noticeable the capsid around the gold nanoparticles.

  1. p5, line 180-181, the authors mention 100% of the encapsidation. The number of counted VLPs in each experiment should be shown as a data.

We appreciate the comment, as we explain in the experimental section, we perform all the encapsidations with an excess of capsid protein CP and because the particles have a highly negatively surface charge, which interact with the highly positively charged N-terminus of the capsid protein CP it is not possible to have “naked” nanoparticles, that is why we consider that an statistic analysis is not needed, also we reassure this by analyzing several TEM images, some of which are in the Supplementary Information file.

  1. p6, line 202, what is the difference between “single capsid” and “multicapsid”?.
  2. p8, line 251, what is “jointed multicapsid”?

ANSWER: We appreciate your question, I think this 2 questions are related so we will answer them together. We describe single capsid and multiwall capsids in line 63 in the new version; icosahedral T=3 capsids are single capsids, while multiwall capsids are formed by one capsid on top of another and so on, forming onion–like structures. Effectively we were using indiscriminately the term multiwall and multishell, so we modify the text by using only multiwall. On the other hand, jointed capsids and multicapsid refer to the same structure that refers to several single capsids “sharing” the same cargo, we modified the text by using only the term jointed capsids.

  1. in p6 line 212, the authors mention most of the AuNS are multishell capsid creating VLPs. How did the authors reveal the multishell capsid structure.
  2. p8, line 256-257, how the authors reveal the encapsidation of AuNSs by two to three protein shells?

ANSWER: We appreciate the questions; we will answer these 2 questions together as they are related. As we mention in the question before, we unified the terms multiwall and multishell when referring to a capsid on top on another. We found that all AuNSs were encapsidated by two to three capsids; this conclusion was determined by TEM micrographs, measuring the thickness of the protein layer around the AuNSs, which is ~10 nm and in the wild-type virus is about 3.5 nm [42], also this experiment was perform in the multiwall formation region as describe in figure 1, and it is described in line 234 in the new version.

  1. The authors should show the data proving the encapsidation of the capsid protein by immunoTEM or ELISA etc.

ANSWER: We appreciate the suggestion, as for detecting that we have protein on the capsids we performed DLS analysis that shows an increase of size, also we analyze the encapsidated nanoparticles under UV-Vis at 280, and lastly in the TEM images the capsids are clearly around the gold nanoparticles, so we consider that those analysis would not give us any extra information for the purposes of this work.

  1. It is doubtful whether the capsid protein formed VLPs or aggregated on the surface of gold nanoparticles.

ANSWER: At those PHs there is a strong protein-protein interaction that will lead to an structured protein formation, it is hard to tell whether the capsid have a perfect T=3 structure on the surface of the nanoparticles or not, in any case a nanoparticle covered in a protein shell can also be define as a VLP, and this will have the same properties.

  1. p8, line 292, “K2CO3” is not correct.

ANSWER: Thank you for noticing, we modified to K2CO3

  1. p10, line 362, “rpm” should not be used.

ANSWER: We appreciate the suggestion; the speed was corrected to 536,600 g in line 335 in the new version

Reviewer 2 Report

The proposed paper shows an interesting approach to encapsidate different types of gold nanoparticles (from spherical to rod-like shapes) within the capsid protein of the plant virus CCMV. The authors observed efficient encapsidation without surface modification of the nanoparticles, just varying the surface charge or the shape.

The paper has some minor weakness, as detailed below:

The introduction is a bit too long, like a review one.

The paper miss to show a stability study over time in the main text. So I suggest author to remove form SI and clearly report and even highlight in the main manusropt how long the encapsidated gold nanostructures are stable and if  there any evidence that Ostwald ripening  has been efficiently prevented with this study.

What about also the biocompatibility of such gold-capsid nanoparticles? Is there any evidence, even deduced from the previous literature, about its safety to human cells?

In Figure 6 empty capsid are not easily recognisable. I suggest the authors to indicate with arrows or coloured circles the various structures obtained (i.e. "A mixture of VLPs with AuNSs cores 209 with CCMV CP and empty capsids of CCMV CP") in figure 6c (as done in the S.I actually)

The efficiency of encapulation  is only evaluated by TEM which is surely  a powerful technique but not statistically significant and showing only localized portion of the sample under examination. I refer to techniques like DLS or Z-potential, showing a prompt change of the sample encapsidated with respect to the pristine (pure gold) one. Figure S3c should be done in principle for all the encapsidated strucutres and reported all in the main manuscript.

In the Method section, authors state "Protein concentration and purity was measured in a UV-Vis spectrophotometer (...), and only the fractions with 280/260 ratios above 1.5 were chosen and used for the encapsidation experiments." How this purification was performed specifically?

other minor comments:

line 38: please review this sentence in  terms of Englih grammar: "in order for the exogenous material to be encapsidated, it has to have similar characteristics as those of the native viral genetic material such as charge and size"

line 58: please correct "and the the capsid"

line 284: please correct "it was possible to encapsidated"

When introducing in the text Figure 4 ad 5 (and also in the respective caption) please indicate at least the technique of characterization used (TEM with negative staining)

In the Method section, please consider to adjust the title of the paragraph of each section or subsection, as only the first is in italic and the other are normal character, with difficulties in decipherying the end of one section and the beginning of the new one.

Author Response

Response to Reviewer 2 Comments

  1. The introduction is a bit too long, like a review one.

ANSWER: We appreciate the comment, as we answered to Reviewer 1, we indeed reduce the introduction by delete the paragraph from 72 to 91 where we explain Figure 1 since it was too repetitive, also, we resume the background from line 113 to 141 in the old version to line 93 to 113 in the new version and we also did some minor changes to complete the Introduction and to make it shorter.

  1. The paper misses to show a stability study over time in the main text. So I suggest author to remove for SI and clearly report and even highlight in the main manusropt how long the encapsidated gold nanostructures are stable and if there any evidence that Ostwald ripening has been efficiently prevented with this study.

ANSWER: We appreciate the comment, in the SI we describe the stability over time of naked AuNS, and it can be seen that the AuNS lack stability. So, we propose that the encapsidation of this type of particles will increase stability as it was shown in reference [44], as it is describe in the main text line 241 to 253 in the new version; a draw back on the use of AuNSs is its well-known instability because in short periods of time they become fragmented due to an Ostwald ripening process [38]. Ostwald ripening leads to the dissolution of smaller nanoparticles into larger ones through coalescence to decrease the interfacial energy between the nanoparticles and the solvent. Colloidal nanoparticles become increasingly unstable with decreasing particle size, due to the relative increase in surface energy. The solubility concentration of the particle is directly proportional to the surface tension and inversely proportional to the radius of the nanoparticle. Therefore, the AuNSs are destroyed by transferring of gold atoms from smaller shells to larger ones. Furthermore, gold nanoshells are susceptible to photothermal fragmentation, when exposed to high IR incident intensities [43]. It has been shown that encapsidation by the CCMV capsid can provide stability against Ostwald ripening in nanolipospheres systems [44]. Therefore, the encapsidation could provide not only a biocompatible surface but also an envelope that can give them a better stability against fragmentation either due to Ostwald ripening or due to photofragmentation as shown in refence [44]. Also, stability measurements of the gold nanoparticles and the encapsidated gold nanoparticles are being performed at the moment, but they will be shown in a future work, when they are finished since this represent a lot of work and time.

  1. What about also the biocompatibility of such gold-capsid nanoparticles? Is there any evidence, even deduced from the previous literature, about its safety to human cells?

ANSWER: Thank you for you questions, there are studies of biocompatibility of nanoparticles [27], and there are also studies of biocompatibility of the CCMV VLPs [57], and in both cases gold nanoparticles and CCMV CP are biocompatible. Therefore, we expect that the encapsidated gold nanoparticles with the CCMV CP will also be biocompatible, reference 57 shows that the VLPs will be able to enter the cell and will not cause damage.

  1. In Figure 6 empty capsid are not easily recognisable. I suggest the authors to indicate with arrows or coloured circles the various structures obtained (i.e. "A mixture of VLPs with AuNSs cores 209 with CCMV CP and empty capsids of CCMV CP") in figure 6c (as done in the S.I actually)

ANSWER: We appreciate the suggestion, the 3 images on figure 6 are from the same experiment, and figure 6c is a lower resolution image, so it is easier to distinguish the empty capsids that are the same size as the wt virus (around 28 nm). We consider that putting arrows in this images will mislead the reader from the main result that we want to show that is that we are able to encapsidate AuNS.

  1. The efficiency of encapsulation is only evaluated by TEM which is surely a powerful technique but not statistically significant and showing only localized portion of the sample under examination. I refer to techniques like DLS or Z-potential, showing a prompt change of the sample encapsidated with respect to the pristine (pure gold) one. Figure S3c should be done in principle for all the encapsidated strucutres and reported all in the main manuscript.

ANSWER: We appreciate the comments, the main purpose of this study is to show that it is possible to encapsidate gold nanoparticles, when report a 100% efficiency due to we perform all the encapsidations with an excess of capsid protein CP and because the particles have a highly negatively surface charge, which interact with the highly positively charged N-terminus of the capsid protein CP it is not possible to have “naked” nanoparticles, that is why we consider that an statistic analysis is not need. Also, we reassure this by analyzing several TEM images.

  1. In the Method section, authors state "Protein concentration and purity was measured in a UV-Vis spectrophotometer (...), and only the fractions with 280/260 ratios above 1.5 were chosen and used for the encapsidation experiments." How this purification was performed specifically?

ANSWER: Thank you for your question, as it is shown in the main text in line 334 to 341 in the new version. It reads: CCMV was harvested and purified from infected Cowpea plants (Vigna ungiculata) following established protocols [4,50]. Briefly, it was dialyzed against disassembly buffer (500 mM CaCl2, 50 mM Tris-HCl pH = 7.5, 1 mM EDTA, 1 mM DTT, and 0.5 mM PMSF) for 24 hours in order to disrupt the viral capsids and separate the CP and the RNA. Then the disassembled virus was ultracentrifuged, at 536,600 g using a TLA110 rotor for 106 min at 4 °C, in order to precipitate the RNA and recover the purified protein still in suspension. The recovered protein was then dialyzed in protein buffer (1 M NaCl, 50 mM Tris-HCl pH = 7.2, 1 mM EDTA, 1 mM DTT, 1 Mm PMSF) that stabilize the protein in dimers [51].

other minor comments:

  1. line 38: please review this sentence in terms of Englih grammar: "in order for the exogenous material to be encapsidated, it has to have similar characteristics as those of the native viral genetic material such as charge and size"

ANSWER: Thank you for your suggestion it was change to: However, for the exogenous material to be encapsidated, must have similar characteristics to the of native viral genetic material, such as charge and size [3].

  1. line 58: please correct "and the the capsid"

ANSWER: Thank you for noticing it was corrected

  1. line 284: please correct "it was possible to encapsidated"

ANSWER: Thank you for noticing it was corrected to: it was possible to encapsidate

  1. When introducing in the text Figure 4 ad 5 (and also in the respective caption) please indicate at least the technique of characterization used (TEM with negative staining)

ANSWER: We appreciate the suggestion, we modified all the captions to include the technique and add a more descriptive title.

  1. In the Method section, please consider to adjust the title of the paragraph of each section or subsection, as only the first is in italic and the other are normal character, with difficulties in decipherying the end of one section and the beginning of the new one.

ANSWER: Thank you very much for your comments, titles were added to each subsection in italics, this was done for the method section as well in the results.

Reviewer 3 Report

In this work, the self-assembly of the capsid protein of cowpea chlorotic mottle virus (CCMV) around spherical gold nanoparticles, gold nanorods and gold nanoshells to form virus-like particles (VLPs) is reported. To induce the protein self-assembly around the negative gold nanoparticles, different pH and ionic strength conditions were used. Finally, the viral capsid protein is study for the encapsidation, showing better biocompatibility, stability and monodispersity.

Minor comments

  1. Materials and Methods should be rewritten, making sections for easier reading.
  2. In general, the legends should contain more information to enable readers to interpret the figures independently.
  3. Results should be rewritten, making sections for easier reading.

Author Response

Response to Reviewer 3 Comments

  1. Materials and Methods should be rewritten, making sections for easier reading.
  2. Results should be rewritten, making sections for easier reading.

ANSWER to comments 1 and 2: We appreciate your suggestions, as Reviewer 2 also suggested, titles were added to each subsection in italics, this for the Methods section as well in the Results.

  1. In general, the legends should contain more information to enable readers to interpret the figures independently.

ANSWER: Thank you for your comment, all the legends were modify giving the reader more information about the characterization method, the conditions of the encapsidation and a more descriptive explanation of each figure.

Round 2

Reviewer 1 Report

Most of doubts have been made clear, but one comment is suggested.

Some experiments are needed to prove the encapsidation of Au nanoparticles with the capsid.

In addition, in the legend of S3, the explanation of (e) is missing.

Author Response

Reviewer 1
  1. Some experiments are needed to prove the encapsidation of Au nanoparticles with the capsid.

Answer: We appreciate the Reviewer´s comment, but it is not clear what type of experiments the Reviewer refers to, but we think it this is not necessary. Images by TEM are well known to probe the encapsidation of different types of cargos. For example, the only two materials used in the encapsidation experiments are the pure capsid protein purified from CCMV wt virus and the gold nanoparticles under a specific buffer. Then we analyze the samples under the transmission electron microscope (TEM). The grids for the TEM images are prepare under the exact same protocol for pure particles with no protein as for encapsidated ones, both with 2% Uranyl acetate as a negative stain, even though the gold particles do not need negative stain due to their high electron density. We add the negative stain to be sure that we are seeing a change due to the protein. It can be seen, for example, in figure S1a for the pure gold nanoparticles that there is no capsid around the black nanoparticle. In contrast, in image S2a for the same type of particles encapsidated with the CCMV CP, the capsids are easily recognizable, all the black particles are surrounded by a white circle that was not there before. Note that there is a good number of empty capsids in the image, due to the excess of protein used in the experiments. The same occurred with the nanoshells as it can be seen in figures S3a and S4a and nanorods in figures S5a and S6a. Thus, TEM images are a good and clear  method that someone can use to elucidate the encapsidation. In addition, we show an analysis of the size of the nanoparticles, before and after the encapsidation. DLS analysis show that an increase in article size corresponding to the thickness of the capsid protein, and is in good agreement with TEM images.

    2. In addition, in the legend of S3, the explanation of (e) is missing.

Answer: Thank you for noticing it, the legend was corrected. It reads now:

e) The arrows show broken gold nanoshells after 20 days due to Ostwald ripening